# Virus-Induced *galactinol-sucrose galactosyltransferase 2* Silencing Delays Tomato Fruit Ripening

**DOI:** 10.3390/plants13182650

**Published:** 2024-09-21

**Authors:** Pengcheng Zhang, Jingjing Wang, Yajie Yang, Jingjing Pan, Xuelian Bai, Ting Zhou, Tongfei Lai

**Affiliations:** College of Life and Environmental Science, Hangzhou Normal University, Hangzhou 310036, China; zpc604@hznu.edu.cn (P.Z.); 2022111010055@stu.hznu.cn (J.W.); 2024111010034@stu.hznu.edu.cn (Y.Y.); 2022111010065@stu.hznu.edu.cn (J.P.); baixl2012@163.com (X.B.)

**Keywords:** tomato fruit, ripening, *galactinol-sucrose galactosyltransferase 2*, virus-induced gene silencing

## Abstract

Tomato fruit ripening is an elaborate genetic trait correlating with significant changes at physiological and biochemical levels. Sugar metabolism plays an important role in this highly orchestrated process and ultimately determines the quality and nutritional value of fruit. However, the mode of molecular regulation is not well understood. Galactinoal-sucrose galactosyltransferase (GSGT), a key enzyme in the biosynthesis of raffinose family oligosaccharides (RFOs), can transfer the galactose unit from 1-α-D-galactosyl-myo-inositol to sucrose and yield raffinose, or catalyze the reverse reaction. In the present study, the expression of *SlGSGT2* was decreased by Potato Virus X (PVX)-mediated gene silencing, which led to an unripe phenotype in tomato fruit. The physiological and biochemical changes induced by *SlGSGT2* silencing suggested that the process of fruit ripening was delayed as well. *SlGSGT2* silencing also led to significant changes in gene expression levels associated with ethylene production, pigment accumulation, and ripening-associated transcription factors (TFs). In addition, the interaction between SlGSGT2 and SlSPL-CNR indicated a possible regulatory mechanism via ripening-related TFs. These findings would contribute to illustrating the biological functions of *GSGT2* in tomato fruit ripening and quality forming.

## 1. Introduction

Tomato (*Solanum lycopersicum*) is an important economic crop and a nutrient and medicinal provider of essential vitamins, minerals, and phytochemicals. Due to many biological advantages such as autogamy, a short life cycle, distinguishable phenotypes, various mutants, genetic transformability, and clear genomic information, tomatoes have been a good model system for fleshy fruit development and ripening studies [1,2,3]. As the climacteric type, ethylene signaling plays a key role in tomato fruit maturity and senescence. Other plant hormones, such as gibberellin, abscisic acid, and auxin, cooperatively participate in this process [4,5,6,7,8]. Transcriptional regulation is another critical regulatory mechanism of fruit ripening [9]. Many transcription factors (TFs) related to fruit ripening have been demonstrated, and they can function by an ethylene-dependent or -independent pathway. Then, a complex regulatory network comprised of TFs has been gradually explored because of functional redundancy, additivity, or harmonious interaction of TFs [10,11,12,13,14]. As important transcriptional and post-transcriptional regulatory factors, noncoding RNAs are also vital in tomato fruit ripening [15,16,17]. Among them, the microRNAs are the most studied class and have gained weight as regulators during tomato fruit development and ripening, such as Sly-miR156a-e, Sly-miR160a, and Sly-miR171a [18,19]. Additionally, emerging reports have indicated that epigenetic modifications such as DNA methylation, demethylation, and histone acetylation are involved in regulating fruit ripening [20,21,22,23]. However, systematic study of control mechanisms at the molecular level is still limited or a matter of debate. New insights are expected to improve understanding of this complex process.

Raffinose is a kind of soluble and non-reducing carbohydrate formed by the galactosyl extension onto the glucose moiety of sucrose. It can be rapidly converted to stachyose by stachyose synthase, and verbascose is then synthesized adding the galactose unit from galactinol to stachyose in a reaction catalyzed by verbascose synthase. Raffinose, stachyose, and verbascose all belong to raffinose family oligosaccharides (RFOs) and are nearly ubiquitous across the plant kingdom [24]. RFOs accumulate in plant fruits or vegetable parts, and the content and composition are associated with genotype, developmental stage, and environmental conditions. Much of the evidence has indicated that RFOs play pivotal roles in plant development, abiotic and biotic stress responses [25,26]. Till now, two pathways (galactinol-dependent or independent) for RFO biosynthesis have been demonstrated. In most plants, the first step is the synthesis of galactinol, catalyzing the galactosyl of UDP-D-galactose transfer to myo-inositol by galactinol synthase. Subsequently, raffinose and myo-inositol are generated by transferring the galactose unit from 1-α-D-galactosyl-myo-inositol to sucrose in a reaction catalyzed by raffinose synthase. Galactinoal-sucrose galactosyltransferase (GSGT) is a raffinose synthase that can catalyze the exchange reaction between raffinose and sucrose [27]. Identification and spatio-temporal expression analysis of raffinose synthase have been performed in many plants, such as peanut, grass species, tea, potato, kiwifruit, cotton, and lentil. Functions of raffinose synthase in plant growth, carbohydrate storage, sugar conversion, seed germination, and tolerance to environmental and biotic stresses have also been revealed [28,29,30,31,32,33,34,35]. In tomato, a total of four *GSGT* genes were predicted, including *SlGSGT*, *SlGSGT-like*, *SlGSGT2,* and *SlGSGT6* (Gene ID: 101257623, 101262300, 101254878, and 101266532). However, it remains elusive whether and how *GSGT* regulates tomato fruit ripening.

Therefore, Potato Virus X (PVX)-induced *SlGSGT2* silencing was performed in the present study. The biochemical and physiological attributes of ripening and non-ripening fruits were assessed. Then, the expression alterations of genes associated with fruit ripening were determined. In addition, the interaction between SlGSGT2 and SlSPL-CNR was evaluated by a yeast two-hybrid assay. The results could contribute to illustrating the biological functions of SlGSGT2 and wider explorations of the regulatory mechanism of tomato fruit ripening.

## 2. Results

### 2.1. The Molecule Structure Features and Expression Profile

SlGSGT2 (UniProtKB: A0A3Q7H4N8; EC:2.4.1.82) belongs to the glycosyl hydrolases 36 family of the alpha-amylase superfamily. It contains 737 amino acids, including 71 negatively charged residues (Asp and Glu) and 51 positively charged residues (Arg and Lys). The molecular weight is 81.44 kDa, and the theoretical pI is 6.20. The predicted secondary structures and hydropathicity of amino acid composition of SlGSGT2 are shown in Appendix A. The number of predicted transmembrane helices (TMHs) in SlGSGT2 is 0. The expected number of amino acids in TMHs and the first 60 amino acids are 3.55 and 3.48, which are both lower than the threshold values (Appendix A). Meanwhile, all known five types of signal peptides are not detected in the SlGSGT2 sequence (Appendix A). In addition, the predicted three-dimensional structure and sequence features of SlGSGT2 are presented in Appendix A.

SlGSGT2 can be detected in the leaf, stem, flower, and root. Of these parts, it has the highest expression level in the leaf, followed by the stem, flower, and root (Figure 1A). At different developmental stages of fruits, the expression of *SlGSGT2* increased gradually from the mature green to the breaker stage, followed by a decline to the red ripening stage (Figure 1B). The ripening-related expression pattern indicated a potential role of *SlGSGT2* in fruit ripening.

Bioinformatic analysis revealed that SlGSGT2 did not contain the typical nuclear-localization signal peptide and TMHs. Using transient transformation of *N. benthamiana* leaves, the subcellular location of the SlGSGT2 protein was determined by the fusion protein SlGSGT2-GFP. The green fluorescence of the fusion protein was observed throughout the cell. The distribution of SlGSGT2-GFP was the same as GFP (Figure 2). The result indicated that SlGSGT2 was widely distributed within the cytoplasm and cell membrane, which was consistent with the reports by Xu et al. (2023) and Schneider and Keller (2009) [35,36].

### 2.2. PVX-Induced SlGSGT2 Silencing Delays Fruit Ripening

Virus-induced gene silencing (VIGS) is an efficient reverse-genetics tool for gene function study in plants. The PVX-based vector has been used to transiently express *SlGSGT2* fragments and silence endogenous *SlGSGT2* in the present study (Figure 3). Through injection of leaf saps from *N. benthamiana* containing PVX or PVX/*pSlGSGT2* transcripts and culturing under standard greenhouse conditions, the fruits showed breaker normally at 37–40 days post anthesis (DPA). After 5 days of breaker, more than 90% of the surface in the AC fruits and AC fruits inoculated with PVX transcripts had turned red. Whereas, about 15% of the fruits inoculated with PVX/*pSlGSGT2* transcripts showed obvious green or orange sectors, which covered about 25–40% of the fruit surfaces. The mesocarp also presented heterogeneous maturity in the cross section (Figure 4A). Meanwhile, compared with red exocarp and mesocarp (REM), the higher expression of PVX/*pSlGSGT2* was detected by RT-PCR in green exocarp and mesocarp (GEM) using a PVX sequencing primer pair (Figure 4B, Appendix A). Then, the effectiveness of PVX-induced *SlGSGT2* silencing was evaluated by qRT-PCT using three detection primer pairs (Appendix A). Compared with REM, the expression of *SlGSGT2* was down-regulated about 30–40% in GEM (Figure 4C). These results indicated that *SlGSGT2* silencing was positively associated with the delayed ripening of tomato fruits.

### 2.3. Effects of SlGSGT2 Silencing on Tomato Ripening-Related Biochemical Characteristics

In general, injection of PVX transcripts has no significant influence on fruit ripening and fruit quality parameters. Through PVX-induced *SlGSGT2* silencing, ripening delay (green sector) and normal ripening (red sector) could be visually inspected on the same fruit (Figure 4). The pH, lycopene content, flavonoid content, and anthocyanin content in GEM were significantly lower than those in REM (Figure 5A,C,E,F) whereas the chlorophyll content was significantly higher in GEM (Figure 5D). There was no significant difference in soluble solids content between REM and GEM (Figure 5B). These were in accordance with the visual changes in physical appearance. The results indicated that PVX-mediated *SlGSGT2* silencing was effective because physiological and biochemical changes were key indicators of fruit ripening.

### 2.4. Effects of SlGSGT2 Silencing on the Expression of Genes Involved in Tomato Fruit Ripening

Ethylene is one of the most important plant hormones and critical for the onset and completion of the fruit ripening process. Due to the typical respiration climacteric behavior of tomato fruit, the biosynthesis of ethylene deserves great attention. Generally, it is largely driven by 1-aminocyclopropanecarboxylic acid synthase (ACS) and 1-aminocyclopropanecarboxylic acid oxidase (ACO). In the present study, compared with REM in the AC fruit inoculated by PVX/*pSlGSGT2* transcripts, the expressions of *SlACS2*, *SlACS4*, *SlACO1*, *SlACO3*, and *SlACO4* were significantly down-regulated in GEM. Whereas, the expression levels of *SlACS1*, *SlACS6*, and *SlACO2* significantly increased in GEM. In addition, the expression level of *SlACS3* has no significant difference between REM and GEM (Figure 6).

TFs are master regulators for the initiation and promotion of tomato fruit ripening [37]. The knowledge concerning the relative expression of TFs is highly valuable for elucidating the effects of *SlGSGT2* silencing on fruit ripening. Compared with REM in AC fruit inoculated by PVX/*pSlGSGT2* transcripts, the expressions of *LeMADS-RIN*, *LeTDR4*, *SlTAGL1*, *LeNAC-NOR*, *SlSPL-CNR*, and *SlMYB12* were significantly down-regulated in GEM whereas the expression levels of *SlAP2a* significantly increased in GEM. In addition, the expression levels of *LeHB1* and *SlMADS1* have no significant difference between REM and GEM (Figure 7).

Carotenoids can function as pigments, nutrients, and the precursors of many important volatile flavor compounds in plants. The content and composition of carotenoids are important indicators of the ripening of tomato fruits [38,39]. In the present study, the relative expression of genes associated with carotenoid biosynthesis in the ripe and non-ripe sectors of *SlGSGT2*-silenced fruit was determined. Compared with REM in AC fruit inoculated by PVX/*pSlGSGT2* transcripts, the expression levels of *PSY1*, *PSY2*, *PSY3*, *PDS*, *ZISO*, *ZDS*, *CRTISO*, *ZEP*, *NCED,* and *β-CRTR* were significantly lower than those in GEM. In addition, *β-LCY* and *ε-LCY* are cyclases that catalyze lycopene to form δ-carotene and γ-carotene. They are negatively correlated with lycopene content in tomato fruit. The expression of *ε-LCY* and *β-LCY* was not affected by *SlGSGT2* silencing (Figure 8).

### 2.5. Potential Interaction Factors of SlGSGT2

In a previous work, we screened the potential interaction proteins of SlSPL-CRN in a tomato fruit cDNA library using the yeast two-hybrid system. One of the original sequences obtained was matched to *SlGSGT2* (XM_010319577.3) and encoded the C-terminal 176 amino acids of SlGSGT2. In the present study, the full-length *SlGSGT2* coding sequence was cloned into pGADT7 and pGBKT7 vectors, as well as *SlSPL-CRN*. In the yeast two-hybrid system with different configurations, the interaction between SlGSGT2 and SlSPL-CRN was confirmed (Figure 9). We also found that the expression of SlGSGT2 was significantly down-regulated in *Cnr* mutants by an iTRAQ analysis [40]. SlSPL-CRN is an important transcription factor and exerts many biological functions during tomato fruit ripening. These guided us to speculate on the relationship between *SlGSGT2* and tomato fruit ripening. In addition, the predicted interaction networks of SlGSGT2 in tomato fruit are shown in Appendix A.

## 3. Discussion

VIGS is one of the rapid and efficient reverse-genetics methods used for gene function analysis in plants [41]. The PVX-mediated gene silencing has been successfully applied in tomato fruit [42,43]. In the present study, the PVX vector harbored a fragment of *SlGSGT2* (456 bp) that was inoculated into tomato fruit and produced a double-stranded RNA copy. It could activate the innate RNAi-mediated defense machinery and lead to the subsequent degradation of the endogenous *SlGSGT2* transcript. As a consequence, the delayed ripening of injected tomato fruits was observed visually. Several biochemical properties, such as pH, pigment content, and flavonoid content, changed synchronously as well. In response to *SlGSGT2* silencing, the expression levels of the main rate-limiting genes *ACS2*, *ACS4,* and *ACO1* for ethylene production were obviously decreased. The expression changes in several known ripening-related TFs genes were also unfavorable for fruit ripening. In addition, due to the significance of carotenoids for fruit color, the down-regulated expression of genes associated with carotenoid biosynthesis was consistent with the unripe phenotype induced by *SlGST2* silencing. These results indicated that PVX-induced *SlGSGT2* silencing could delay the normal ripening of tomato fruit.

GSGT can determine the levels of RFOs in higher plants by transferring a galactosyl group of galactinol to sucrose or catalyzing the reverse reaction. Besides the roles in plant abiotic stress tolerance and biotic stress response, RFOs can regulate plant growth and development by functioning as transport carbohydrates in the phloem, storage compounds in sink tissues, and soluble metabolites [25]. During seed maturation and germination in *Cicer arietinum*, RFOs were likely required to maintain a steady-state level of reducing sugars [44]. The germination of *RS4/RS5* double knock-out *Arabidopsis thaliana* seeds was delayed for five days because of the absence of RFOs [45]. Similarly, pea seeds had a significantly lower germination rate when the breakdown of RFOs was blocked by a galactosidase inhibitor [46]. Hybrid poplar overexpressing the *Arabidopsis thaliana* galactinol synthase gene *AtGolS3* possessed higher cellulose and xylem starch contents and lower lignin content. Then, cell wall development was disturbed, and the tension wood was formed [47]. In cucumber, two alkaline a-galactosidase genes, *CsAGA1* and *CsAGA2*, played important but partly different roles in the hydrolysis of RFOs [48]. Further study suggested that *CsAGA2* could affect fruit set and development by regulating sugar communication between source activity and sink strength [49]. Therefore, we speculated that RFO content changes caused by *GSTS2* silencing may trigger a series of metabolic changes, ultimately impacting sugar transport, cell differentiation, and the ripening process of tomato fruit.

The catabolism of RFOs is as critical as the biosynthesis reaction. RFOs can be hydrolyzed to D-galactose and sucrose by alkaline α-galactosidase and acid α-galactosidase. Sucrose can be degraded into UDP-glucose, glucose, and fructose by sucrose synthase or invertase. They participate in various biosynthetic and metabolic processes in the vacuole, apoplast, or cytosol [27]. Sugars not only serve as energy sources that fuel organ and tissue growth, but the composition and content of sugars also determine the nutritional quality and sweetness of fruit [50]. In addition, sugars can function as signaling molecules that coordinate plant development and growth by regulating gene expression [51,52]. For example, the sugar-inducible protein kinase VsSK1 could promote the expression of monosaccharide transporter genes during the ripening stage in grapes [53]. Similarly, the sugar-inducible transcription factor SUSIWM1 and Tonoplast Sugar Transporter ClAGA2 have a positive correlation with the sweetness of watermelon fruit [54,55]. Likewise, the higher sugar content could increase capabilities for addressing the oxidative processes associated with fruit ripening in plum [56]. Qin et al. (2016) provided solid evidence for the roles of sugar metabolism in promoting fruit ripening [57]. Therefore, the content balance of sugars and RFOs in tomato fruit may be disturbed by *GSGT2 silencing* in this study, which contributed to the delay of tomato fruit ripening.

Furthermore, galactinol was the galactose donor in the raffinose biosynthetic process [58]. Many reports have shown that galactinol served as a signaling component with many regulatory functions. It could initiate metabolic changes or regulate the transcription of genes to improve stress tolerance [59,60]. Zhang et al. (2024) found that a decrease in galactinol content would prompt color transformation and ethylene release in tomato fruit by regulating the expression of genes involved in carotenoid and ethylene metabolism [61]. Conversely, the increase in galactinol content would inhibit tomato fruit ripening. In the present study, *SlGSGT2* silencing might block the biosynthesis of raffinose, induce the accumulation of galactinol, and lead to the inhibition of tomato fruit ripening.

It is worth noting that GSGT2 can interact with 1R-MYB TF in chickpeas [62]. In the previous study, we found that the fragment of *SlGSGT2* encoding peptides can interact with SlSPL-CNR bait protein in the screening of a fruit cDNA library. Then, the interaction between SlSPL-CNR and SlGSGT2 with full length was confirmed by the yeast two-hybrid analysis in this study. SlSPL-CNR is a multifunctional TF, and the *Cnr* epimutation generates a severe unripe phenotype with a colorless mealy pericarp [63,64]. Compared with wildtype, the expression of SlGSGT2 protein was down-regulated during fruit ripening in *Cnr* mutants [40]. Meanwhile, SlSPL-CNR can interact with sucrose non-fermenting related kinase 1 (SnRK1) to affect tomato fruit ripening [43]. SnRK1 is a serine/threonine kinase that plays a crucial role in plants. It can control the expressions of a large number of genes involved in the adjustment of growth t, the reprogramming of metabolism, and the stress response [65,66]. In cucumber calli, SnRK1 could influence the RFOs metabolism by regulating three α-galactosidase genes [67]. Therefore, the tight connections of SlGSGT2, SlSPL-CNR, and SlSnRK1 in space may provide one useful hint for exploring the function mechanism of SlGSGT2 in the tomato fruit ripening process.

Therefore, we speculated that *SlGSGT2* silencing may disturb the balance of sucrose, raffinose, and galactinol contents, affect sugars and RFOs signaling mediated hormonal and metabolic networks, influence the transcription regulation of key TFs, and finally inhibit the fruit ripening process.

## 4. Materials and Methods

### 4.1. Plant Materials and Growth Conditions

Tomato (*Solanum lycopersicum* cv. Ailsa Craig) and tobacco (*Nicotiana benthamiana*) were grown in a greenhouse using standard culture practices (25 °C and 80% humidity, 16 h light–8 h dark of photoperiod) with regular additions of fertilizer and water. Flowers were tagged at anthesis. Developmental and ripening stages of fruits were recorded as days post anthesis (DPA) and days post breaker (DPB). Breaker was defined as fruits with the color change from green to yellow.

### 4.2. RNA Extraction and qRT-PCR

The harvested roots, stems, leaves, or fruits at different ripening stages were ground into powder in liquid nitrogen. Total RNAs of samples were extracted by RNeasy Plant Mini Kit (Qiagen, Hilden, Germany). The first-strand cDNA was produced using the Fast Quant RT Kit (Tiangen, Beijing, China). The quantitative real-time PCR detection (qRT-PCR) was performed using a 2×Ultra SYBR mixture in a CFX96-Real Time System (Bio-Rad, Hercules, CA, USA). The PCR program was as below: 95 °C (10 min), followed by 40 circles of 95 °C (15 s), 58 °C (15 s), and 72 °C (20 s). The intensity change in SYBR Green fluorescence and the threshold cycle (Ct) over the background were calculated for each reaction. Samples were normalized utilizing 18S rRNA and *glyceraldehyde 3-phosphate dehydrogenase* (*GAPDH*), respectively. The relative expression levels were measured using the 2^(−ΔΔCt)^ analysis method and presented as the average results based on two reference genes. All specific primer sequences for genes involved in ethylene production, ripening-associated transcript factors, and carotenoid biosynthesis are listed in Appendix A.

### 4.3. Subcellular Localization of SlGSGT2

Subcellular localization of SlGSGT2 was determined as described by Lai et al. (2020) [43]. Briefly, the coding sequence of *SlGSGT2* was cloned into the pCAMBIA1300/35S:sGFP vector to generate the 35S:SlGSGT2-sGFP transient expression vector. The indicated protein was transiently expressed in tobacco leaves by Agrobacterium (strain GV3101)-mediated infiltration. After infiltration, tobacco epidermal leaf cells were subjected to confocal laser scanning microscopy analysis by Zeiss LSM 900 (Carl Zeiss AG, Oberkochen, Germany). The fluorescence was observed at an excitation wavelength of 497 nm and an emission wavelength of 517 nm.

### 4.4. Bioinformatic Prediction of SlGSGT2

The SOPMA (Self Optimized Prediction Method with Alignment) (Prabi, Lyon, France) was used to predict the secondary structure of SlGSGT2. ProtScale (Expasy, SIB, Lausanne, Switzerland) was used to compute the hydropathicity profile of SlGSGT2. DeepTMHMM V1.0 (DTU Health Tech, Lyngby, Denmark) was used to detect and predict the transmembrane topology of SlGSGT2. The SignalP V6.0 (DTU Health Tech, Lyngby, Denmark) was used to predict signal peptides of SlGSGT2. STRING V12.0 (SIB, Lausanne, Switzerland) was used to determine known and predicted protein–protein interactions of SlGSGT2. The fully automated protein structure homology-modeling software SWISS-MODEL V1.0 (SIB and Biozentrum, Basel, Switzerland) was used to predict the three-dimensional structure of SlGSGT2.

### 4.5. PVX-Induced Genes Silencing

The fragments of *SlGSGT2* (Position: 211-666/XM_010319577.3) containing 456 bp were amplified by RT-PCR from fruit cDNA (30 DPA) using the PrimerSTAR HS DNA polymerase (Takara, Osaka, Japan). The PCR products were digested by restriction enzymes *Cla* I and *Eag* I and cloned into the PVX vector. Nucleotide sequencing was performed to confirm the constructs by using a pair of PVX specific primers. Then, the right constructs were linearized by restriction enzyme *Spe* I (NEB, Ipswich, MA, USA), purified by the High Pure PCR Product Purification Kit (Roche, Basel, Switzerland), and suspended in RNase-free water with 1 unit/μL RNasin Ribonuclease Inhibitor (Promega, Madison, Wisconsin, USA). Single-strand RNA transcripts were synthesized using the Riboprobe In Vitro Transcription System (Promega, Madison, WI, US) and mechanically inoculated on young leaves of *N. benthamiana*. After 10 days, systemically infected leaves were harvested and freeze-dried in a freeze dryer (FreeZone 2.5 Plus, LABCONCO, Kansas City, MO, USA). Subsequently, approximately 0.1 g of leaf tissue was ground in 2 mL of TE buffer (pH 7.5) and needle-injected into the carpopodium of immature tomato fruit (about 15 PDA). At 5 DPB, fruits with obvious non-ripening sectors were photographed. The exocarp and mesocarp in red or green sectors were separated and stored at −80 °C.

### 4.6. Determination of Physiological and Biochemical Parameters

The pH value of fruit flesh was evaluated utilizing a 3520 pH Meter (JENWAY, Staffordshire, UK) in 15 mL of diluted juice from a 5 g sample. The soluble solids content was determined utilizing a refractometer (LB20T, Suwei, Shanghai, China) following the product instruction. For lycopene content detection, the fruit flesh (10 g) was grounded in liquid nitrogen and repeatedly washed by methanol until the supernatant was colorless. Then, total lycopene was extracted by 50 mL chloroform, and the absorbance of the supernatant was measured at 485 nm. The standard curve equation was as follows: y = 0.2764x + 0.0013, R^2^ = 0.9954. The lycopene content was represented by mg per kg of fresh weight. The total chlorophyll content and the anthocyanin content were assessed as described by Wang et al. (2009) and Zhang et al. (2014), respectively [68,69]. The flavonoid content was determined using the aluminum chloride colorimetric assay [70].

### 4.7. Yeast Two-Hybrid Assay

The interaction of SlGSGT2 and SlSPL-CNR was determined by the Matchmaker Gold Yeast Two-Hybrid System (Takara, Osaka, Japan) following the product manual. The *SlSPL-CNR* and *SlGSGT2* coding sequences were amplified and cloned into the pGBKT7 or pGADT7 vectors. The construction strategy is represented in Figure 9B. The purified pGBKT7/SlSPL-CNR and pGADT7/SlGSGT2 (or pGBKT7/SlGSGT2 and pGADT7/SlSPL-CNR) vectors were co-transformed into yeast strain AH109 utilizing Yeastmaker Yeast Transformation System 2 (Takara, Osaka, Japan). Then, the suspension of the transformation products was spread on YPDA medium, SD/–Leu/–Trp double dropout (DDO) medium, or SD/–Ade/–His/–Leu/–Trp quadruple dropout (QDO) medium. After 3–5 d of culturing at 30 °C, positive colonies were isolated from the QDO medium, and the plasmids were extracted using the TIANprep Yeast Plasmid DNA Kit (Tiangen, Beijing, China). The inserted fragments on the vectors were amplified and sequenced. AH109/pGBKT7-53+pGADT7-T was taken as a positive control. AH109, AH109/pGADT7+pGBKT7, AH109/pGADT7-SlGSGT2+pGBKT7, and AH109/pGADT7+pGBKT7-SlGSGT2 were employed as negative controls. The primer pairs used in the yeast two-hybrid assay are listed in Appendix A.

### 4.8. Statistical Analysis

Data were collected from at least three independent experiments, and statistical analyses were performed by Excel software (Microsoft Office 2013, Redmond, Washington, USA). Analysis of variance (ANOVA) was used to compare more than two means. Mean separations were analyzed using Duncan’s multiple range test. Differences at *p* < 0.05 were considered to be significant.

## 5. Conclusions

In conclusion, the PVX-induced *SlGSGT2* silencing could lead to an unripe phenotype in tomato fruit. The changes in gene expression levels related to ethylene production, pigment accumulation, and ripening-associated TFs confirmed the delay of fruit ripening, as well as the physiological and biochemical characteristics. In addition, the interaction between SlGSGT2 and SlSPL-CNR indicated a possible regulatory mechanism via ripening-related TFs. These findings would contribute to illustrating the biological functions of GSGT2 in tomato fruit ripening and quality-forming.

## Figures and Tables

**Figure 1 plants-13-02650-f001:**
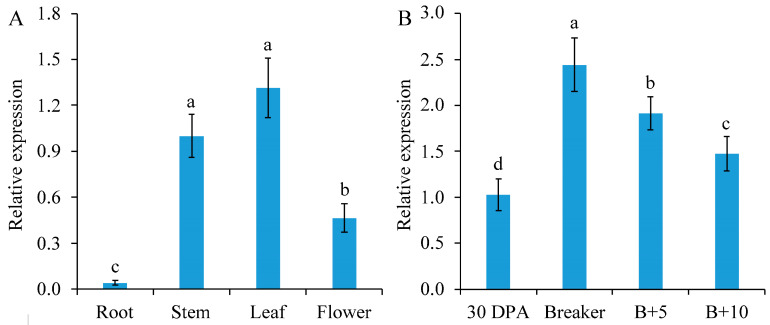
Developmental expression of *SlGSGT2* in different parts of plants (**A**) or different ripening stages of tomato cultivar Ailsa Craig fruit (**B**). The values represent the means of three biological replicates, and the bars represent the standard deviation of the means. Lowercase letters indicate significant differences at *p* < 0.05. DPA: days post anthesis; B+5: 5 days after breaker; B+10: 10 days after breaker.

**Figure 2 plants-13-02650-f002:**
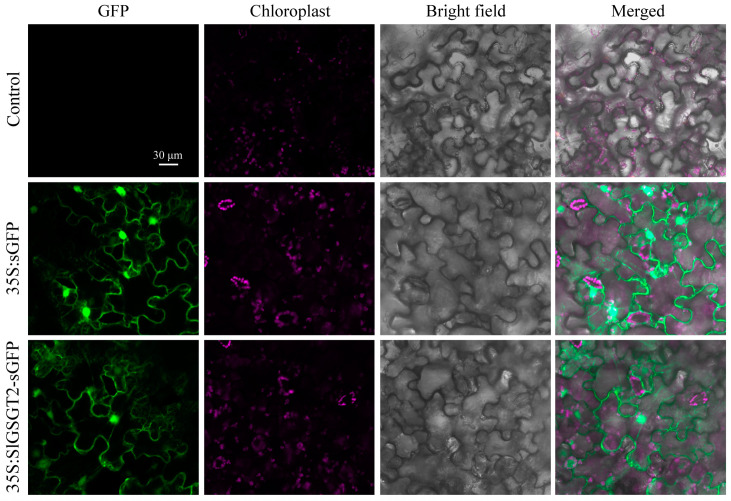
Characterization of the nuclear localization signal for GSGT2-sGFP in *N. benthamiana* leaf cells. Leaves infiltrated with Agrobacterium (strain GV3101, Tiangen, Beijing, China) serve as controls. Leaves are taken on the seventh day post-inoculation and examined under a confocal microscope.

**Figure 3 plants-13-02650-f003:**
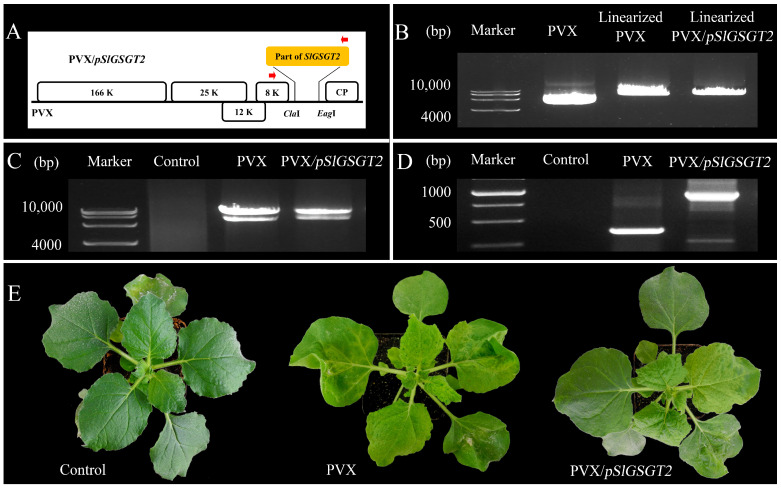
In vitro transcription of PVX/*pSlGSGT2*. (**A**) The schematic representation of the PVX/*pSlGSGT2* vector. (**B**) The linearization of PVX and PVX*/pSlGSGT2* vectors by a restriction endonuclease *Spe* I. (**C**) The in vitro transcription RNA products of PVX and PVX/*pSlGSGT2*. (**D**) RT-PCR detection of PVX and PVX/*pSlGSGT2* in infected leaves of *N. benthamiana*. (**E**) The phenotypes of *N. benthamiana* with PVX and PVX/*pSlGSGT2* through mechanical inoculation after 10 days of culturing. The plant is mock inoculated with H_2_O as the negative control.

**Figure 4 plants-13-02650-f004:**
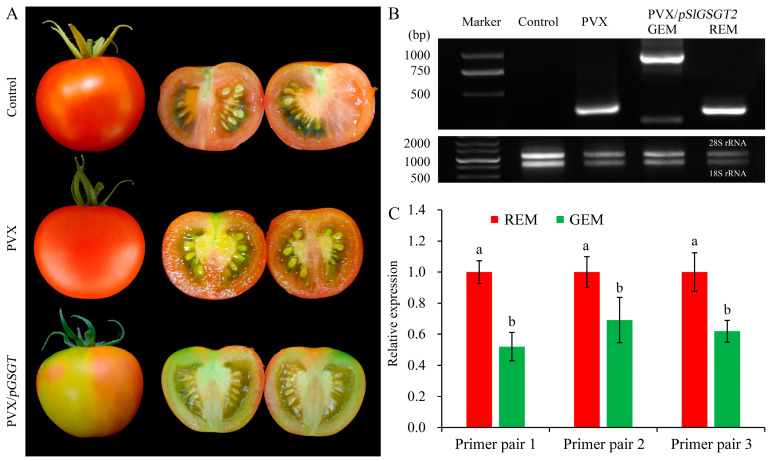
PVX-induced *SlGSGT2* silencing delays tomato fruit ripening. (**A**) Ripening phenotypes of wild-type AC fruit and fruit injected with PVX or PVX/*pGSGT2* at 5 days after breaker. (**B**) RT-PCR detection of PVX and PVX/*pGSGT2* in tomato fruit at 5 days after breaker. (**C**) The relative expression of *SlGSGT2* in REM and GEM of tomato fruit injected with PVX/*pGSGT2* at 5 days after breaker. Three primer pairs are designed to detect the expression level of *SlGSGT2* by qRT-PCR. The primer information is listed in Appendix A. The values represent the means of three biological replicates, and the bars represent the standard deviation of the means. Lowercase letters indicate significant differences at *p* < 0.05. REM: red exocarp and mesocarp; GEM: green exocarp and mesocarp.

**Figure 5 plants-13-02650-f005:**
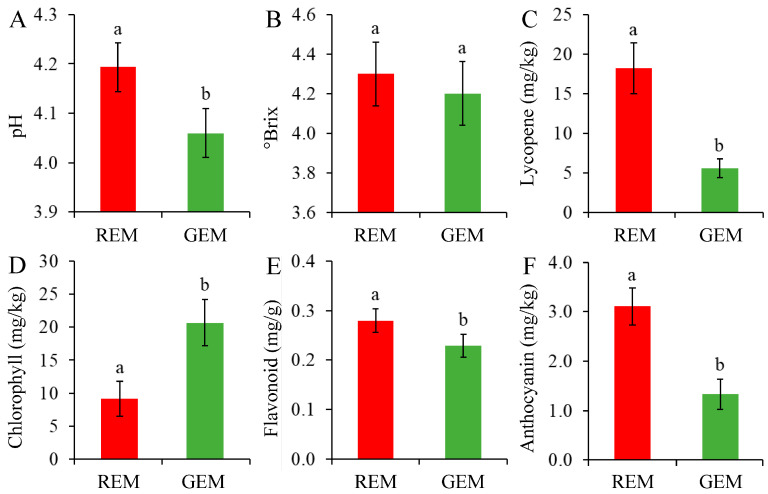
Biochemical characteristics of the ripe and non-ripe sectors of *SlGSGT2*-silenced fruit at 5 days after breaker. (**A**) pH, (**B**) soluble solids content, (**C**) lycopene content, (**D**) chlorophyll content, (**E**) flavonoid content, and (**F**) anthocyanin content. The values represent the means of three biological replicates, and the bars represent the standard deviation of the means. Lowercase letters indicate significant differences at *p* < 0.05. REM: red exocarp and mesocarp; GEM: green exocarp and mesocarp.

**Figure 6 plants-13-02650-f006:**
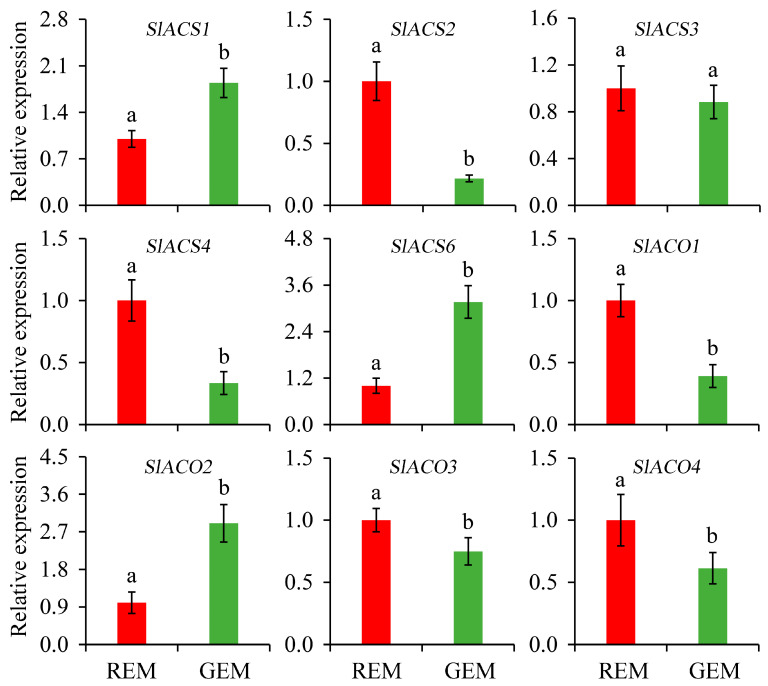
Relative expression of genes related to ethylene biosynthesis in the ripe and non-ripe sectors of *SlGSGT2*-silenced fruit at 5 days after breaker. The values represent the means of three biological replicates, and the bars represent the standard deviation of the means. Lowercase letters indicate significant differences at *p* < 0.05. REM: red exocarp and mesocarp; GEM: green exocarp and mesocarp; ACS: 1-aminocyclopropanecarboxylic acid synthase; ACO: 1-aminocyclopropanecarboxylic acid oxidase.

**Figure 7 plants-13-02650-f007:**
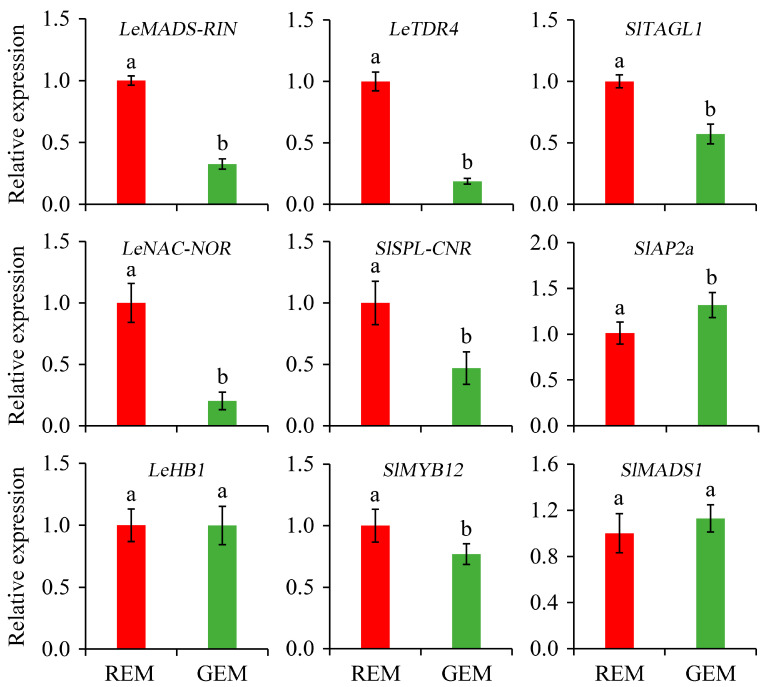
Relative expression of ripening-associated transcription factor genes in the ripe and non-ripe sectors of *SlGSGT2*-silenced fruit at 5 days after breaker. The values represent the means of three biological replicates, and the bars represent the standard deviation of the means. Lowercase letters indicate significant differences at *p* < 0.05. REM: red exocarp and mesocarp; GEM: green exocarp and mesocarp.

**Figure 8 plants-13-02650-f008:**
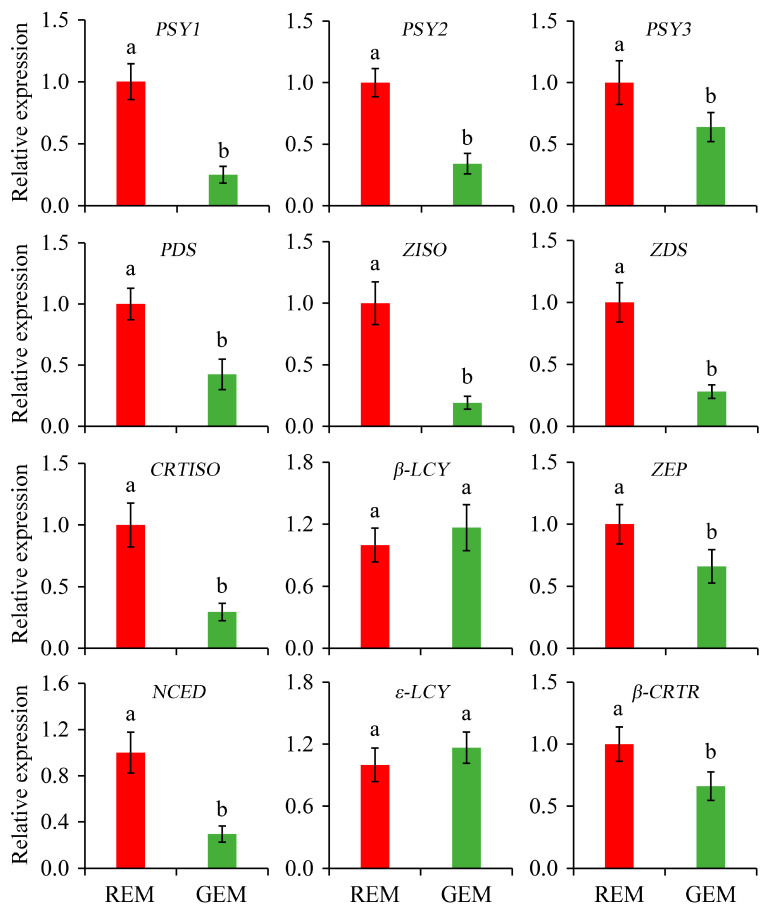
Relative expression of genes associated with carotenoid biosynthesis in the ripe and non-ripe sectors of *SlGSGT2*-silenced fruit at 5 days after breaker. The values represent the means of three biological replicates, and the bars represent the standard deviation of the means. Lowercase letters indicate significant differences at *p* < 0.05. REM: red exocarp and mesocarp; GEM: green exocarp and mesocarp; PSY: Phytoene synthase; PDS: Phytoene desaturase; ZISO: Zeta-carotene isomerase; ZDS: Zeta-carotene desaturase; CRTISO: Carotenoid isomerase; β-LCY: Lycopene β-cyclase; ZEP: Zeaxanthin epoxidase; NCED: 9-cis-epoxycarotenoid dioxygenase; ε-LCY: Lycopene ε-cyclase; β-CRTR: β-carotene hydroxylase.

**Figure 9 plants-13-02650-f009:**
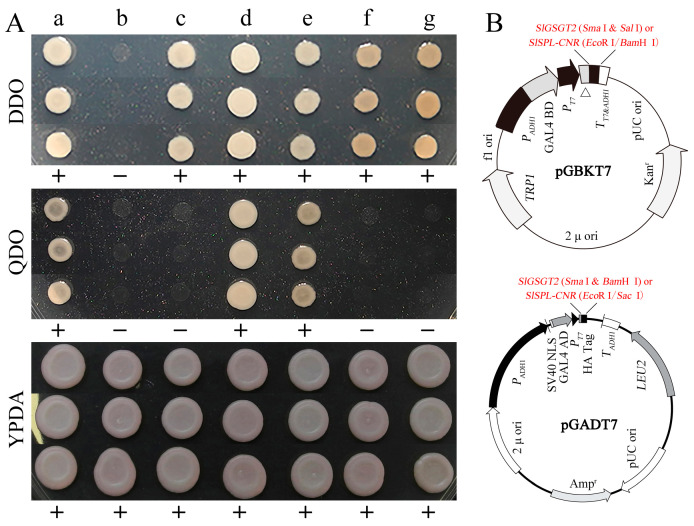
Interaction between SlSPL-CNR and SlGSGT2. (**A**) Interaction between SlSPL-CNR and SlGSGT2 in the yeast two-hybrid system. DDO: SD/–Leu/–Trp double dropout; QDO: SD/–Ade/–His/–Leu/–Trp quadruple dropout; YPDA: yeast extract peptone dextrose adenine medium. a: positive control AH109/pGBKT7-53+pGADT7-T; b: negative control AH109; c: AH109/pGADT7+ pGBKT7; d: AH109/pGADT7-SlSPL-CNR+ pGBKT7-SlGSGT2; e: AH109/pGADT7- SlGSGT2+ pGBKT7-SlSPL-CNR; f: AH109/pGADT7-SlGSGT2+ pGBKT7; g: AH109/pGADT7+ pGBKT7-SlGSGT2. (**B**) The schematic representation of pGBKT7 and pGADT7 derived vectors used in the yeast two-hybrid system. Detailed vector information of pGADT7 or pGBKT7 can be found in the product manual.

## Data Availability

Data are contained within the article and Appendix A.

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
