# Peer review of "Virus-Induced galactinol-sucrose galactosyltransferase 2 Silencing Delays Tomato Fruit Ripening"

_plants, 2024, doi:10.3390/plants13182650_

Round 1

Reviewer 1 Report

Comments and Suggestions for Authors

Dear Authors,

I have an opportunity to review manuscript entitled: ”Virus-induced galactinol-sucrose galactosyltransferase 2 silencing delays tomato fruit ripening” submitted to Plants MDPI.

Authors concentrated on  potato virus (PVX) mediated Solanum lycopersicum galactinoal-sucrose galactosyltransferase (GSGT) gene silencing, which has influence on phenotype in tomato fruit.

The main suggestion is that taking into account that sugar metabolism plays an important role in this highly orchestrated process and finally determines the quality and nutritional value of tomato fruits in which GSGT participated, sugar profiles changes should be added- sucrose, raffinose and galactinol contents should be measure in control as well as in silenced tomato plants.

First of all, Authors presented relative gene expression- why only one reference gene were taking into account- this analyses should be repeated and statistically analyzed for two reference genes (Please, be aware, the statistical error / standard deviations are very high in current form);

Moreover, Authors stated that “Developmental expression of SlGSGT2 in different tissues”- for sure leaf, stem, flower are not plant tissues !-Please, correct it;

Furthermore, resolution for “localization signal for GSGT2-GFP” is too small- especially when Authors underlines nuclear localization;-It should be changed;

Figure 3 and 4 are very good constructed in a good quality- but in my opinion instead of “red and green sector” the fruit tissues should be added;

All keywords should be added in full name not in abbreviations;

Please, check carefully if all gene names are in italics;

I suggest extensive English language corrections, because in some places the Authors statements are difficult to understand;

Authors should enhanced conclusions which in current form are added only as a speculation form- for example: “We speculated SlGSGT2 silencing may disturb the balance of sucrose, raffinose and galactinol contents, affect sugars and RFOs signaling mediated hormonal and metabolic network, then inhibit the fruit ripening process”;

Figure 10-String analyses can be added to supplementary files;

Comments on the Quality of English Language

Please, make extensive English language corrections by native speaker, because some Authors statements are difficult to understand.

Author Response

Dear Ms. Lea Tao,

We received your E-mail with the reviews of our manuscript (NO.: Plants-3184446) entitled “Virus-induced galactinol-sucrose galactosyltransferase 2 silencing delays tomato fruit ripening” on August 28, 2024. First of all, I apologize for the submission delay because it takes us more time to carry out the necessary experiments. We would like to thank you and two reviewers very much for giving us such good comments about this paper. We have earnestly revised this paper, according to the comments and suggestions made by you and two reviewers. The specific revisions have been highlighted in bright yellow in the revised manuscript. A detailed explanation of how we have dealt with the points raised by the reviewers can be found in attachment.

Finally, we would like to thank you and two reviewers again and we are looking forward to hearing your reply.

Best wishes!

Yours sincerely,

Tongfei Lai

On behalf of all the authors

College of Life and Environmental Science, Hangzhou Normal University, Hangzhou 310036, China

Reviewer 2 Report

Comments and Suggestions for Authors

Dear Editor,

I went thoroughly on the manuscript ‘Virus-induced galactinol-sucrose galactosyltransferase 2 silencing delays tomato fruit ripening’. The authors were explained well about the role of SlGSGT2 in delaying the ripening of tomato. However, I have few questions on their finding on this particular gene silencing and this effect on ripening. My queries mentioned below.

1. I found various typo errors and spelling mistakes in this manuscript. For instance, LN-72, It should be Potato Virus X. 

LN-118, It should be qRT-PCR

LN-549; Fig-3, ‘with’ and ‘through’ should not be italics.

2. In Fig 4: Which gene primers used to run RT-PCR for the PVX

3. How much length of SlGSGT2 sequence used for the silencing experiments?

4. Is there any comparing studies with TRV-silencing vector?

5. Did find any non-specific silencing of this gene? If not how come you confirmed the silencing is gene specific?

6. Did you infiltrate the plants with GV3101 cells as a control?

Comments on the Quality of English Language

I found errors in English language. Please proof it once again.

Author Response

(The authors gave the same response as above.)

Round 2

Reviewer 1 Report

Comments and Suggestions for Authors

Authors significantly improved manuscript - especially results part are improved;

Almost all suggestion are considered- conclusions are "strenghten", substantial errors are corrected. I also take into accont and understand Authors explanation obout sugar content analyses.

Reviewer 2 Report

Comments and Suggestions for Authors

Dear Authors,

I really appreciate your responses for my quires. Now I felt that this manuscript is suitable for acceptance.